# Transfer among Agents: An Efficient Multiagent Transfer Learning Framework

## Abstract

Transfer Learning has shown great potential to enhance the single-agent Reinforcement Learning (RL) efficiency, by sharing learned policies of previous tasks. Similarly, in multiagent settings, the learning performance can also be promoted if agents can share knowledge with each other. However, it remains an open question of how an agent should learn from other agents' knowledge. In this paper, we propose a novel multiagent option-based policy transfer (MAOPT) framework to improve multiagent learning efficiency. Our framework learns what advice to give to each agent and when to terminate it by modeling multiagent policy transfer as the option learning problem. MAOPT provides different kinds of variants which can be classified into two types in terms of the experience used during training. One type is the MAOPT with the Global Option Advisor which has the access to the global information of the environment. However, in many realistic scenarios, we can only obtain each agent's local information due to partial observation. The other type contains MAOPT with the Local Option Advisor and MAOPT with the Successor Representation Option (SRO) which are suitable for this setting and collect each agent's local experience for the update. In many cases, each agent's experience is inconsistent with each other which causes the option-value estimation to oscillate and to become inaccurate. SRO is used to handle the experience inconsistency by decoupling the dynamics of the environment from the rewards to learn the option-value function under each agent's preference. MAOPT can be easily combined with existing deep RL approaches. Experimental results show it significantly boosts the performance of existing deep RL methods in both discrete and continuous state spaces.

## 1 Introduction

Transfer Learning has shown great potential to accelerate single-agent RL via leveraging prior knowledge from past learned policies of relevant tasks (Yin & Pan, 2017; Yang et al., 2020). Inspired by this, transfer learning in multiagent reinforcement learning (MARL) (Claus & Boutilier, 1998; Hu & Wellman, 1998; Bu et al., 2008; Hernandez-Leal et al., 2019; da Silva & Costa, 2019) is also studied with two major directions: 1) transferring knowledge across different but similar MARL tasks and 2) transferring knowledge among multiple agents in the same MARL task. For the former, several works explicitly compute similarities between states or temporal abstractions (Hu et al., 2015; Boutsioukis et al., 2011; Didi & Nitschke, 2016) to transfer across similar tasks with the same number of agents, or design new network structures to transfer across tasks with different numbers of agents (Agarwal et al., 2019; Wang et al., 2020). In this paper, we focus on the latter direction due to the following intuition: in a multiagent system (MAS), each agent's experience is different, so the states each agent encounters (the degree of familiarity to the different regions of the whole environment) are also different; if we figure out some principled ways to transfer knowledge across different agents, all agents could form a big picture about the MAS even without exploring the whole space of the environment, and this will definitely facilitate more efficient MARL (da Silva et al., 2020).

Transferring knowledge among multiple agents is still investigated at an initial stage, and the assumptions and designs of some recent methods are usually simple. For example, LeCTR (Omidshafiei et al., 2019) and HMAT (Kim et al., 2020) adopted the teacher-student framework to learn to teach by assigning each agent two roles (i.e., the teacher and the student), so the agent could learn

when and what to advise other agents or receive advice from other agents. However, both LeCTR and HMAT only consider two-agent scenarios. Liang & Li (2020) proposed a method under the teacher-student framework where each agent asks for advice from other agents through learning an attentional teacher selector. However, they simply used the difference of two unbounded value functions as the reward signal which may cause instability. DVM (Wadhwania et al., 2019) and LTCR Xue et al. (2020) are two proposed multiagent policy distillation methods to transfer knowledge among more than two agents. However, both methods decompose the solution into several stages in a coarse-grained manner. Moreover, they consider the distillation equally throughout the whole training process, which is counter-intuitive. A good transfer should be adaptive rather than being equally treated, e.g., the transfer should be more frequent at the beginning of the training since agents are less knowledgeable about the environment, while decay as the training process continues because agents are familiar with the environment gradually and should focus more on their own knowledge.

In this paper, we propose a novel MultiAgent Option-based Policy Transfer (MAOPT) framework which models the policy transfer among multiple agents as an option learning problem. In contrast to the previous teacher-student framework and policy distillation framework, MAOPT is adaptive and applicable to scenarios consisting of more than two agents. Specifically, MAOPT adaptively selects a suitable policy for each agent as the advised policy, which is used as a complementary optimization objective of each agent. MAOPT also uses the termination probability as a performance indicator to determine whether the advice should be terminated to avoid negative transfer. Furthermore, to facilitate the scalability and robustness, MAOPT contains two types: one type is MAOPT with the global option advisor (MAOPT-GOA), the other type consists of MAOPT with the local option advisor (MAOPT-LOA) and MAOPT with the successor representation option advisor (MAOPT-SRO). Ideally, we can obtain the global information to estimate the option-value function, where MAOPT-GOA is used to select a joint policy set, in which each policy is advised to each agent. However, in many realistic scenarios, we can only obtain each agents' local experience, where we adopt MAOPT-LOA and MAOPT-SRO. Each agent's experience may be inconsistent due to partial observations, which may cause the inaccuracy in option-value's estimation. MAOPT-SRO is used to overcome the inconsistency in multiple agents' experience by decoupling the dynamics of the environment from the rewards to learn the option-value function under each agent's preference. MAOPT can be easily incorporated into existing DRL approaches and experimental results show that it significantly boosts the performance of existing DRL approaches both in discrete and continuous state spaces.

## 2 PRELIMINARIES

Stochastic Games (Littman, 1994) are a natural multiagent extension of Markov Decision Processes (MDPs), which model the dynamic interactions among multiple agents. Considering the fact agents may not have access to the complete environmental information, we follow previous work's settings and model the multiagent learning problems as partially observable stochastic games (Hansen et al., 2004). A Partially Observable Stochastic Game (POSG) is defined as a tuple $\langle \mathcal{N}, \mathcal{S}, \mathcal{A}^1, \cdots, \mathcal{A}^n, \mathcal{T}, \mathcal{R}^1, \cdots, \mathcal{R}^n, \mathcal{O}^1, \cdots, \mathcal{O}^n \rangle$, where $\mathcal{N}$ is the set of agents; $\mathcal{S}$ is the set of states; $\mathcal{A}^i$ is the set of actions available to agent $i$ (the joint action space $\mathcal{A} = \mathcal{A}^1 \times \mathcal{A}^2 \times \cdots \times \mathcal{A}^n$); $\mathcal{T}$ is the transition function that defines transition probabilities between global states: $\mathcal{S} \times \mathcal{A} \times \mathcal{S} \to [0, 1]$; $\mathcal{R}^i$ is the reward function for agent $i$: $\mathcal{S} \times \mathcal{A} \to \mathbb{R}$ and $\mathcal{O}^i$ is the set of observations for agent $i$. A policy $\pi^i$: $\mathcal{O}^i \times \mathcal{A}^i \to [0, 1]$ specifies the probability distribution over the action space of agent $i$. The goal of agent $i$ is to learn a policy $\pi^i$ that maximizes the expected return with a discount factor $\gamma$: $J = \mathbb{E}_{\pi^i} \left[ \sum_{t=0}^{\infty} \gamma^t r_t^i \right]$.

**The Options Framework.** Sutton et al. (1999) firstly formalized the idea of temporally extended action as an option. An option $\omega \in \Omega$ is defined as a triple $\{\mathcal{I}_\omega, \pi_\omega, \beta_\omega\}$ in which $\mathcal{I}_\omega \subset \mathcal{S}$ is an initiation state set, $\pi_\omega$ is an intra-option policy and $\beta_\omega : \mathcal{I}_\omega \to [0, 1]$ is a termination function that specifies the probability an option $\omega$ terminates at state $s \in \mathcal{I}_\omega$. An MDP endowed with a set of options becomes a Semi-Markov Decision Process (Semi-MDP), which has a corresponding optimal option-value function over options learned using intra-option learning. The options framework considers the *call-and-return* option execution model, in which an agent picks an option $o$ according to its option-value function $Q_\omega(s, \omega)$, and follows the intra-option policy $\pi_\omega$ until termination, then selects a next option and repeats the procedure.

**Deep Successor Representation (DSR).** The successor representation (SR) (Dayan, 1993) is a basic scheme that describes the state value function by a prediction about the future occurrence of all states under a fixed policy. SR decouples the dynamics of the environment from the rewards. Given a transition $(s, a, s', r)$, SR is defined as the expected discounted future state occupancy:

$$M(s, s', a) = \mathbb{E}\left[\sum_{t=0}^{\infty} \gamma^t \mathbb{1}[s_t = s'] | s_0 = s, a_0 = a\right], \tag{1}$$

where $\mathbb{1}[.]$ is an indicator function with value of one when the argument is true and zero otherwise. Given the SR, the Q-value for selecting action $a$ at state $s$ can be formulated as the inner product of the SR and the immediate reward: $Q^\pi(s, a) = \sum_{s' \in \mathcal{S}} M(s, s', a) \mathcal{R}(s')$.

DSR (Kulkarni et al., 2016) extends SR by approximating it using neural networks. Specifically, each state $s$ is represented by a $D$-dimensional feature vector $\phi_s$, which is the output of the network parameterized by $\theta$. Given $\phi_s$, SR is represented as $m_{sr}(\phi_s, a|\tau)$ parameterized by $\tau$, a decoder $g_{\bar{\theta}}(\phi_s)$ parameterized by $\bar{\theta}$ outputs the input reconstruction $\hat{s}$, and the immediate reward at state $s$ is approximated as a linear function of $\phi_s$: $\mathcal{R}(s) \approx \phi_s \cdot \mathbf{w}$, where $\mathbf{w} \in \mathbb{R}^D$ is the weight vector. In this way, the Q-value function can be approximated by putting these two parts together as: $Q^\pi(s, a) \approx m_{sr}(\phi_s, a|\tau) \cdot \mathbf{w}$. The stochastic gradient descent is used to update parameters $(\theta, \tau, \mathbf{w}, \bar{\theta})$. Specifically, the loss function of $\tau$ is:

$$L(\tau, \theta) = \mathbb{E}\left[(\phi_s + \gamma m'_{sr}(\phi_{s'}, a'|\tau') - m_{sr}(\phi_s, a|\tau))^2\right], \tag{2}$$

where $a' = \arg\max_a m_{sr}(\phi'_s, a) \cdot \mathbf{w}$, and $m'_{sr}$ is the target SR network parameterized by $\tau'$ which follows DQN (Mnih et al., 2015) for stable training. The reward weight $\mathbf{w}$ is updated by minimizing the loss function: $L(\mathbf{w}, \theta) = (\mathcal{R}(s) - \phi_s \cdot \mathbf{w})^2$. The parameter $\bar{\theta}$ is updated using an L2 loss: $L(\bar{\theta}, \theta) = (\hat{s} - s)^2$. Thus, the loss function of DSR is the composition of the three loss functions: $L(\theta, \tau, \mathbf{w}, \bar{\theta}) = L(\tau, \theta) + L(\mathbf{w}, \theta) + L(\bar{\theta}, \theta)$.

## 3 MULTIAGENT OPTION-BASED POLICY TRANSFER (MAOPT)

### 3.1 FRAMEWORK OVERVIEW

In this section, we describe our MAOPT framework in detail. Figure 1 illustrates the MAOPT framework which contains $n$ agents interacting with the environment and corresponding option advisors. At each step, each agent $i$ obtains its own observation $o^i$, selects an action $a^i$ following its policy $\pi^i$, and receives its reward $r^i$. Each option advisor initializes the option set, and selects an option for each agent. During the training phase, the option advisor uses samples from all agents to update the option-value function and corresponding termination probabilities. Each agent is advised by an option advisor, and the advice is to exploit this advised policy through imitation, which serves as a complementary optimization objective (each agent does not know which policy it imitates and how the extra loss function is calculated)*. The

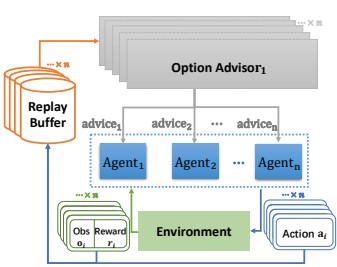

Figure 1: Framework overview.

exploitation of this advised policy is terminated as the selected option terminates and then another option is selected. In this way, each agent efficiently exploits useful information from other agents and as a result, the learning process of the whole system is accelerated and improved. Note that in the following section we assume the agents using the option advisor are homogeneous, i.e., agents share the same option set. While our MAOPT can also support the situation where each agent is initialized with different numbers of options, e.g., each agent only needs to imitate its neighbours. To achieve this, instead of input states into the option-value network, we just input the pair of states and options to the network and output a single option-value.

Our proposed MAOPT can be classified into two types in terms of the experience used during training. One type is MAOPT with the global option advisor (MAOPT-GOA) which has the access to the

---

*We provide the theoretical analysis to show this objective ensures to converge to an improved policy and will not affect the convergence of the original RL algorithm.

global information (i.e., $(s, \vec{a}, \mathbf{r}, s')$, where $\mathbf{r} = \sum_{i=1}^{n} r^i$) of the environment. Thus, MAOPT-GOA selects a joint option as the advice set given the global observation of the environment and then evaluates the performance of the selected joint option. Selecting a joint option means that each advice given to each agent begins and ends simultaneously. However, in many realistic scenarios, we can only obtain each agent's local information due to the partial observation. Moreover, the degree of familiarity to the environment of each agent is different, then some agents may need to imitate their teachers for a longer time. Therefore, a more flexible way to control each advice when to terminate individually is necessary. The other type contains MAOPT with the local option advisor (MAOPT-LOA), and MAOPT with the successor representation option advisor (MAOPT-SRO) which collects each agent's local experience for the update. In many cases, each agent's experience is inconsistent with each other, e.g., each agent has an individual goal to achieve or has different roles, and the rewards assigned to each agent are different. If we simply use all experiences for the update, the option-value estimation would oscillate and become inaccurate. MAOPT-SRO is used to handle the experience inconsistency by decoupling the dynamics of the environment from the rewards to learn the option-value function under each agent's preference.

## 3.2 MAOPT-GOA

In cases where we have access to the global information of the environment, the global option advisor is used to advise each agent. The procedure of MAOPT-GOA is described as follows (pseudo-code is included in the appendix). First, MAOPT-GOA initializes the joint option set $\Omega^1 \times \Omega^2 \times \cdots \times \Omega^n$ (where $\Omega^i = \{\omega^1, \cdots, \omega^n\}$). Each option $\omega^i$ corresponds to agent $i$'s policy $\pi^i$. The joint option-value function is defined as $Q_{\vec{\omega}}(s, \vec{\omega}|\psi)$ parameterized by $\psi$ which evaluates the performance of each joint option $\vec{\omega}$. The corresponding target network is parameterized by $\psi'$ which copies from $\psi$ every $k$ steps. The termination network parameterized by $\varpi$ outputs the termination probability $\beta(s', \vec{\omega}|\varpi)$ of the joint option $\vec{\omega}$.

The update of the joint option-value network update follows previous work (Sutton et al., 1999; Bacon et al., 2017). We first samples $B$ transitions uniformly from the global replay buffer, for each sample $(s, \vec{a}, \mathbf{r}, s')$, we calculate the joint $U$ function, the joint option-value function upon arrival:

$$U(s', \vec{\omega}|\psi') = (1 - \beta(s', \vec{\omega}|\varpi)) Q_{\vec{\omega}}(s', \vec{\omega}|\psi') + \beta(s', \vec{\omega}|\varpi) \max_{\vec{\omega}' \in \vec{\Omega}} Q_{\vec{\omega}}(s', \vec{\omega}'|\psi'). \qquad (3)$$

Then, the option-value network minimizes the following loss:

$$L_{\vec{\omega}} = \frac{1}{B} \sum_b \left( \mathbf{r_b} + \gamma U(s_{b+1}, \vec{\omega}|\psi') - Q_{\vec{\omega}}(s_b, \vec{\omega}|\psi) \right)^2. \qquad (4)$$

where $\mathbf{r_b} = \sum_n r_b^i$.

According to the call-and-return option execution model, the termination probability $\beta_{\vec{\omega}}$ controls when to terminate the selected joint option and then to select another joint option accordingly, which is updated w.r.t $\varpi$ as follows (Bacon et al., 2017):

$$\varpi = \varpi - \alpha_{\varpi} \frac{\partial \beta(s', \vec{\omega}|\varpi)}{\partial \varpi} A(s', \vec{\omega}|\psi') + \xi, \qquad (5)$$

where, $A(s', \vec{\omega}|\psi')$ is the advantage function which can be approximated as $Q_{\vec{\omega}}(s', \vec{\omega}|\psi') - \max_{\vec{\omega}' \in \vec{\Omega}} Q_{\vec{\omega}}(s', \vec{\omega}'|\psi')$, and $\xi$ is a regularization term to ensure explorations (Bacon et al., 2017). Then, given each state $s$, MAOPT-GOA selects a joint option $\vec{\omega}$ following the $\epsilon$-greedy strategy over the joint option-value function. Then MAOPT-GOA calculates the cross-entropy $H(\pi_{\omega}|\pi^i)$ between each intra-option policy $\pi_{\omega}$ and each agent's policy $\pi^i$, and gives it to each agent respectively, serving as a complementary optimization objective of each agent, which means that apart from maximizing the cumulate reward, the agent also imitates the intra-option policy $\pi_{\omega}$ by minimizing the loss function $L_{tr}^i$. The imitation for the intra-option policy is terminated as the option terminated, and then another option is selected to provide advice for the agent. The formula of the loss function $L_{tr}^i$ is given as follows:

$$L_{tr}^i = f(t) H(\pi_{\omega}|\pi^i), \qquad (6)$$

where, $f(t) = 0.5 + \tanh(3 - \mu t)/2$ is the weighting factor of $H(\pi_{\omega}|\pi^i)$. $\mu$ is a hyper-parameter that controls the decrease degree of the weight. This means that at the beginning of learning, each agent exploits knowledge from other agents mostly. As learning continues, knowledge from other

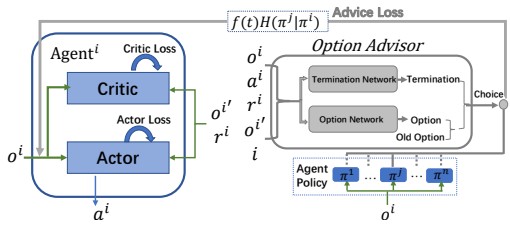

Figure 2: MAOPT-LOA.

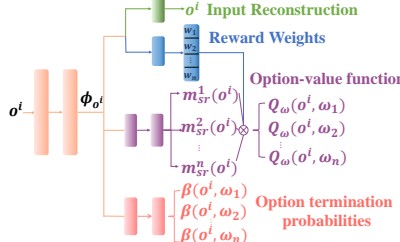

Figure 3: The SRO architecture.

agents becomes less useful and each agent focuses more on the current self-learned policy. Note that we cannot calculate the cross-entropy between the discrete action policies directly. To remedy this, we apply the softmax function with some temperature to the discrete action vectors to transform the actions into discrete categorical distributions.

In MAOPT-GOA, each advice given to each agent begins and ends simultaneously. While for each agent, when to terminate the reusing of other agents' knowledge should be decided asynchronously and individually since the degree of familiarity to the environment of each agent is probably not identical. Moreover, in many realistic scenarios, we can only obtain each agent's local information due to partial observation. Therefore, a more flexible way to advise each agent is necessary. In the following section, we describe the second type of MAOPT in detail.

## 3.3 MAOPT-LOA

MAOPT-LOA equips each agent an option advisor, and each advisor uses local information from all agents for estimation. How MAOPT-LOA is applied in actor-critic methods is illustrated in Figure 2 (pseudo-code is included in the appendix). Firstly, MAOPT-LOA initializes $n$ options $\Omega = \{\omega^1, \omega^2, \cdots, \omega^n\}$. Each option $\omega^i$ corresponds to agent $i$'s policy $\pi^i$. The input of option network parameterized by $\psi$ and termination network parameterized by $\varpi$ is the local observation $o^i$ of each agent $i$. The option-value function $Q_\omega(o^i, \omega|\psi)$ and termination probability $\beta(o^i, \omega|\varpi)$ are used to evaluate the performance of each option $\omega^i \in \Omega$.

The update of the option-value function and the termination probability is similar to that in MAOPA-GOA. For the update of each agent $i$, MAOPT-LOA first selects an option $\omega$ from $\{\omega^1, \omega^2, \cdots, \omega^n\}$ following $\epsilon$-greedy strategy over the option-value function. Then MAOPT-LOA calculates the cross-entropy $H(\pi_\omega|\pi^i)$ between each intra-option policy $\pi_\omega$ and each agent's policy $\pi^i$, and gives it to each agent respectively, contributing to a complementary loss function $L_{tr}^i$ for each agent.

Note that the option-value network and termination network collect experience from all agents for the update. What if the experience from one agent is inconsistent with others? In a POSG, each agent can only obtain the local observation and individual reward signal, which may be different for different agents even at the same state. If we use inconsistent experiences to update one shared option-value network and termination network, the estimation of the option-value function would oscillate and become inaccurate. We propose MAOPT-SRO to address this problem. MAOPT-SRO decouples the dynamics of the environment from the rewards to learn the option-value function under each agent's preference. In this way, MAOPT-SRO can address such sample inconsistency and learn the option-value and the corresponding termination probabilities under each agent's preference which is described in the next section.

## 3.4 MAOPT-SRO

MAOPT-SRO applies a novel option learning algorithm, Successor Representation Option (SRO) learning to learn the option-value function under each agent's preference. The SRO network architecture is shown in Figure 3, with each observation $o^i$ from each agent $i$ as input. $o^i$ is input through two fully-connected layers to generate the state embedding $\phi_{o^i}$, which is transmitted to three network sub-modules. The first sub-module contains the state reconstruction model which ensures $\phi_{o^i}$ well representing $o^i$, and the weights for the immediate reward approximation at local observation $o^i$. The immediate reward is approximated as a linear function of $\phi_{o^i}$: $\mathcal{R}^i(\phi_{o^i}) \approx \phi_{o^i} \cdot \mathbf{w}$,

where $\mathbf{w} \in \mathbb{R}^D$ is the weight vector. The second sub-module is used to approximate SR for options $m_{sr}(\phi_{o^i}, \omega|\tau)$ which describes the expected discounted future state occupancy of executing the option $\omega$. The corresponding target network is parameterized by $\tau'$ which copies from $\tau$ every $k$ steps. The last sub-module is used to update the termination probability $\beta(\phi_{o_{i'}}, \omega|\varpi)$, which is similar to that in MAOPT-LOA described in Section 3.3.

---

**Algorithm 1** MAOPT-SRO.

---

1: Initialize: option set $\Omega = \{\omega^1, \omega^2, \cdots, \omega^n\}$, state feature parameters $\theta$, reward weights $\mathbf{w}$, state reconstruction network parameters $\bar{\theta}$, termination network parameters $\varpi$, SR network parameters $\tau$, SR target network parameters $\tau'$, batch size $T$ for PPO, replay buffer $\mathcal{D}^i$, actor network parameters $\rho^i$, and critic network parameters $\upsilon^i$ for each agent $i$
2: **for** each episode **do**
3:     Start from state $s$
4:     **for** each agent $i$ **do**
5:         Select an option $\omega$
6:         Select an action $a^i \sim \pi^i(o^i)$
7:     **end for**
8:     Perform the joint action $\vec{a} = \{a^1, \cdots, a^n\}$
9:     Observe reward $\vec{r} = \{r^1, \cdots, r^n\}$ and new state $s'$
10:     **for** each agent $i$ **do**
11:         Store transition $(o^i, a^i, r^i, o^{i'}, \omega, i)$ to replay buffer $\mathcal{D}^i$
12:         Select another option $\omega'$ if $\omega$ terminates
13:     **end for**
14:     **for** every $T$ steps **do**
15:         **for** each agent $i$ **do**
16:             Set $\pi_{old}^i = \pi^i$
17:             Calculate the advantage $A^i = \sum_{t' > t} \gamma^{t'-t} r_t^i - V_{\upsilon^i}(o_t^i)$
18:             Optimize the critic loss $L_c^i = -\sum_{t=1}^T (\sum_{t' > t} \gamma^{t'-t} r_t^i - V_{\upsilon^i}(o_t^i))^2$
19:             The option advisor calculates the transfer loss $L_{tr}^i = f(t) H(\pi_\omega | \pi^i)$
20:             Optimize the actor loss $\bar{L}_a^i = \sum_{t=1}^T \frac{\pi^i(a_t^i|o_t^i)}{\pi_{old}^i(a_t^i|o_t^i)} A_t^i - \lambda KL[\pi_{old}^i|\pi^i] + L_{tr}^i$ w.r.t $\rho^i$
21:         **end for**
22:     **end for**
23:     Sample a batch of $B/N$ transitions $(o^i, a^i, r^i, o^{i'}, i)$ from each $\mathcal{D}^i$
24:     Optimize $L(\bar{\theta}, \theta) = \left(g_{\bar{\theta}}(\phi_{o^i}) - o^i\right)^2$ w.r.t $\bar{\theta}, \theta$
25:     Optimize $L(\mathbf{w}, \theta) = \left(r^i - \phi_{o^i} \cdot \mathbf{w}\right)^2$ w.r.t $\mathbf{w}, \theta$
26:     **for** each $\omega$ **do**
27:         **if** $\pi_\omega$ selects action $a^i$ at observation $o^i$ **then**
28:             Calculate $\tilde{U}(\phi_{o_{i'}}, \omega|\tau')$
29:             Set $y \leftarrow \phi_{o^i} + \gamma\tilde{U}(\phi_{o_{i'}}, \omega|\tau')$
30:             Optimize the following loss w.r.t $\tau$: $L(\tau, \theta) = \frac{1}{B} \sum_b (y_b - m_{sr}(\phi_{o^i}, \omega|\tau))^2$
31:             Optimize the termination network w.r.t $\varpi$: $\varpi = \varpi - \alpha_\varpi \frac{\partial \beta(\phi_{o_{i'}}, \omega|\varpi)}{\partial \varpi} A(\phi_{o_{i'}}, \omega|\tau') + \xi$
32:         **end if**
33:     **end for**
34:     Copy $\tau$ to $\tau'$ every $k$ steps
35: **end for**

---

Given $m_{sr}(\phi_{o^i}, \omega|\tau)$, the SRO-value function can be approximated as: $Q_\omega(\phi_{o^i}, \omega) \approx m_{sr}(\phi_{o^i}, \omega|\tau) \cdot \mathbf{w}$. Since options are temporal abstractions (Sutton et al., 1999), SRO also needs to calculate the $\tilde{U}$ function which is served as SR upon arrival, indicating the expected discounted future state occupancy of executing an option $\omega$ upon entering a state. Given the transition $(o^i, a^i, r^i, o^{i'})$, we firstly introduce the SR upon arrival $\tilde{U}$ as follows:

$$\tilde{U}(\phi_{o_{i'}}, \omega|\tau') = (1 - \beta(\phi_{o_{i'}}, \omega|\varpi))m_{sr}(\phi_{o_{i'}}, \omega|\tau') + \beta(\phi_{o_{i'}}, \omega|\varpi)m_{sr}(\phi_{o_{i'}}, \omega'|\tau'), \quad (7)$$

where $\omega' = \arg\max_{\omega \in \Omega} m_{sr}(\phi_{o_{i'}}, \omega|\tau') \cdot \mathbf{w}$.

We consider MAOPT-SRO combing with PPO (Schulman et al., 2017), a popular single-agent policy-based RL. The way MAOPT-SRO combines with other policy-based RL algorithms is similar. Algorithm 1 illustrates the whole procedure of MAOPT-SRO. First, we initialize the network parameters for the state embedding network, reward prediction network, state reconstruction network, termination network, SR network, SR target network, and the actor and critic networks of each agent (Line 1). For each episode, each agent $i$ first obtains its local observation $o^i$ which corresponds to the current state $s$ (Line 3). Then, MAOPT-SRO selects an option $\omega$ for each agent $i$ (Line 5), and each agent selects an action $a^i$ following its policy $\pi^i$ (Line 6). The joint action $\vec{a}$ is performed, then the reward $\mathbf{r}$ and new state $s'$ is returned from the environment (Lines 8,9). The transition is stored in the replay buffer $\mathcal{D}^i$ (Line 11). If $\omega$ terminates, MAOPT-SRO selects another option for agent $i$ (Line 12). For each update step, each agent updates its critic network by minimizing the loss $L_c^i$ (Line 18), where $T$ is the length of the trajectory segment. Then each agent updates its actor network by minimizing the summation of the original loss and the transfer loss $L_{tr}^i$ (Line 20). For the update of SRO, it first samples a batch of $B/N$ transitions from each agent's buffer $\mathcal{D}^i$, which means there are $B$ transitions in total for the update (Line 23). SRO loss is composed of three components: the state reconstruction loss $L(\bar{\theta}, \theta)$, the loss for reward weights $L(\mathbf{w}, \theta)$ and SR loss $L(\tau, \theta)$. The state reconstruction network is updated by minimizing two losses $L(\bar{\theta}, \theta)$ (Line 24) and $L(\mathbf{w}, \theta)$ (Line 25). The second sub-module, SR network approximates SR for options and is updated by minimizing the standard L2 loss $L(\tau, \theta)$ (Lines 26-30). At last, the termination probability of the selection option is updated (Line 31), where $A(\phi_{o^{i'}}, \omega | \tau')$ is the advantage function and approximated as $m_{sr}(\phi_{o^{i'}}, \omega | \tau') \cdot \mathbf{w} - \max_{\omega \in \Omega} m_{sr}(\phi_{o^{i'}}, \omega | \tau') \cdot \mathbf{w}$, and $\xi$ is a regularization term to ensure explorations.

## 4 EXPERIMENTAL RESULTS

In this section, we evaluate the performance of MAOPT combined with the common single-agent RL algorithm (PPO (Schulman et al., 2017)) and MARL algorithm (MADDPG (Lowe et al., 2017)). The test domains include two representative multiagent games, Pac-Man and multiagent particle environment (MPE) (illustrated in the appendix). Specifically, we first combine MAOPT-LOA and MAOPT-SRO with PPO on Pac-Man to validate whether MAOPT-SRO successfully solves the sample inconsistency due to the partial observation. Then, we combine MAOPT-GOA, MAOPT-LOA, and MAOPT-SRO with two baselines (MADDPG and PPO) on MPE to further validate whether MAOPT-SRO is a more flexible way for knowledge transfer among agents and enhances the advantage of our framework. We also compare with DVM (Wadhwania et al., 2019), which is the most recent multiagent transfer method[†].

### 4.1 PAC-MAN

Pac-Man (van der Ouderaa, 2016) is a mixed cooperative-competitive maze game with one pac-man player and two ghost players. The goal of the pac-man player is to eat as many pills as possible and avoid the pursuit of ghost players. For ghost players, they aim to capture the pac-man player as soon as possible. In our settings, we aim to control the two ghost players and the pac-man player as the opponent is controlled by well pre-trained PPO policy. The game ends when one ghost catches the pac-man player or the episode exceeds 100 steps. Each player receives $-0.01$ penalty each step and $+5$ reward for catching the pac-man player.

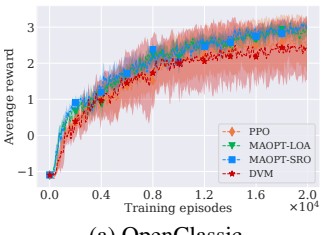

(a) OpenClassic

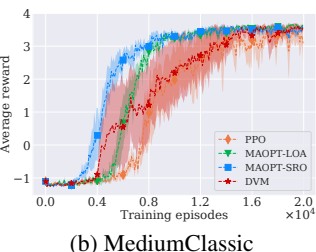

(b) MediumClassic

Figure 4: The performance on Pac-Man.

We consider two Pac-Man scenarios (OpenClassic and MediumClassic) with the game difficulties increasing. Figure 4 (a) presents the average rewards on the OpenClassic scenario. We can see that both MAOPT-LOA and MAOPT-SRO perform better than other methods and achieve the average

---

[†]The details of neural network structures and parameter settings are in the appendix, and we share network parameters among all homogeneous agents (Gupta et al., 2017; Rashid et al., 2018).

discount rewards of $+3$ approximately with smaller variance. In contrast, PPO and DVM only achieve the average discount rewards of $+2.5$ approximately with larger variance. This phenomenon indicates that both MAOPT-LOA and MAOPT-SRO enable efficient knowledge transfer between two ghosts, thus facilitating better performance.

Next, we consider a more complex Pac-Man game scenario, where the layout size is larger than the former and it contains obstacles (walls). From Figure 4 (b) we can observe that the advantage of MAOPT-LOA and MAOPT-SRO is much more obvious compared with PPO and DVM. Furthermore, MAOPT-SRO performs best among all methods, which means that MAOPT-SRO effectively selects more suitable advice for each agent. The reason that MAOPT performs better than DVM is that MAOPT enables each agent to effectively exploit useful information from other agents through the option-based call-and-return mechanism, which successfully avoids negative transfer when other agents' policies are only partially useful. However, DVM just transfers all information from other agents through policy distillation. By comparing the results of the two scenarios, we see that the superior advantage of MAOPT-SRO increases when faced with more challenging scenarios. Intuitively, as the environmental difficulties increase, agents are harder to explore the environment and to learn the optimal policy. In such a case, agents need to exploit the knowledge of other agents more efficiently, which would greatly accelerate the learning process as demonstrated by MAOPT-LOA and MAOPT-SRO.

## 4.2 MPE

MPE (Lowe et al., 2017) is a simple multiagent particle world with continuous observation and discrete action space. We evaluate the performance of MAOPT on two scenarios: predator and prey, and cooperative navigation. The predator and prey contains three agents which are slower and want to catch one adversary (rewarded $+10$ by each hit). The adversary is faster and wants to avoid being hit by the other three agents. Obstacles block the way. The cooperative navigation contains four agents, and four corresponding landmarks. Agents are penalized with a reward of $-1$ if they collide with other agents. Thus, agents have to learn to cover all the landmarks while avoiding collisions. Both games end when exceeding 100 steps.

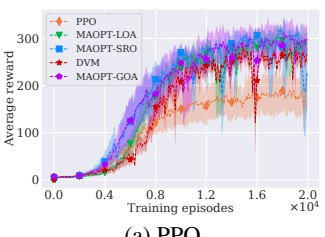

(a) PPO

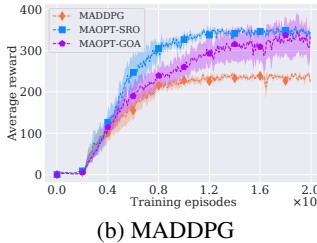

(b) MADDPG

Figure 5: The performance on predator and prey.

Both domains contain the sample inconsistency problem since each agent's local observation contains the relative distance between other agents, obstacles, and landmarks. Moreover, in cooperative navigation, each agent is assigned a different task, i.e., approaching a different landmark from others, which means each agent may receive different rewards under the same observation. Therefore, we cannot directly use all experience to update one shared option-value network. In such a case, we design an individual option learning module for each agent in MAOPT-LOA, which only collects one agent's experience to update the option-value function.

Figure 5 (a) shows the average rewards on predator and prey. We can see that all our proposed MAOPT-GOA, MAOPT-LOA, and MAOPT-SRO (combined with PPO) achieve higher average rewards than PPO and DVM. Figure 5 (b) demonstrates a similar phenomenon that both MAOPT-GOA and MAOPT-SRO (combined with MADDPG) perform better than vanilla MADDPG, and MAOPT-SRO performs best among all methods. This is because MAOPT-SRO uses all agents' experience for the update and efficiently distinguishes which part of the information is useful and provides positive advice for each agent. Furthermore, it uses the individual termination probability to determine when to terminate each agent's advice, which is a more flexible manner, thus facilitating more efficient and effective knowledge transfer among agents.

Table 1 shows the average distance between each agent and its nearest landmark (line 1), and the average collision frequencies of agents (line 2) in cooperative navigation. In this game, agents are required to cover all landmarks while avoiding collisions. Therefore, a better result means to get a closer average distance between agents and landmarks, and less collision frequencies among agents.

Table 1: Average of collisions and average distance from a landmark in cooperative navigation.

|  | PPO | DVM | MADDPG | MAOPT-GOA | MAOPT-LOA | MAOPT-SRO |
|---|---|---|---|---|---|---|
| Avg. dist. | 1.802 | 1.685 | 1.767 | 1.446 | 1.476 | **1.366** |
| Collisions | 0.163 | 0.144 | 0.209 | 0.153 | 0.124 | **0.122** |

We can see that MAOPT-GOA, MAOPT-LOA, and MAOPT-SRO achieve the less collisions and the shorter average distance from landmarks than other methods. Furthermore, MAOPT-SRO performs best among all methods. The superior advantage of MAOPT is due to the effectiveness in identifying the useful information from other agents' policies. Therefore, each agent exploits useful knowledge of other agents and as a result, thus leading to the least collisions and the minimum distance from landmarks.

Finally, we provide an ablation study to investigate whether MAOPT-SRO selects a suitable policy for each agent, thus efficiently enabling agents to exploit useful information from others. Figure 6 presents the action movement in the environment, each arrow is the direction of movement caused by the specific action at each location. Four figures show the direction of movement caused by the action selected from the policy of an agent at $t_1 = 6 \times 10^5$ steps (Figure 6(a), top left), and at $t_2 = 2 \times 10^6$ (Figure 6(c), bottom left); the direction of movement caused by the action selected from the intra-option policies of MAOPT-SRO at $t_1 = 6 \times 10^5$ steps (Figure 6(b), top right), and at $t_2 = 2 \times 10^6$ steps (Figure 6(d), bottom right) respectively. The preferred direction of movement should be towards the blue circle. We can see that actions selected by the intra-option policies of MAOPT-SRO are more accurate than those selected from the agent's own policy, namely, more prone to pursue the adversary (blue). This indicates that the advised policy selected by MAOPT-SRO performs better than the agent itself, which means MAOPT-SRO successfully distinguishes useful knowledge from other agents. Therefore, the agent can learn faster and better after exploiting knowledge from this advised policy than learning from scratch.

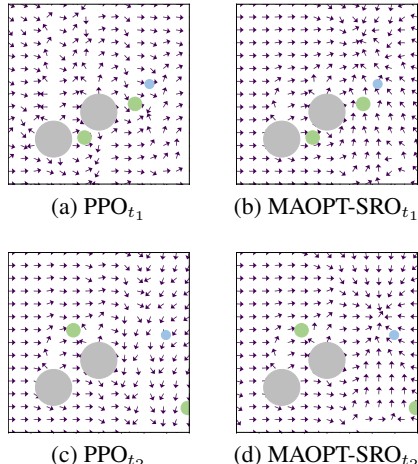

(a) $\text{PPO}_{t_1}$     (b) $\text{MAOPT-SRO}_{t_1}$

(c) $\text{PPO}_{t_2}$     (d) $\text{MAOPT-SRO}_{t_2}$

Figure 6: Analysis of agent 1's policy and MAOPT-SRO's policy.

## 5 CONCLUSION AND FUTURE WORK

In this paper, we propose a novel MultiAgent Option-based Policy Transfer (MAOPT) framework for efficient multiagent learning by taking advantage of option-based policy transfer. Our framework learns what advice to give to each agent and when to terminate it by modeling multiagent transfer as the option learning problem. Furthermore, to facilitate the robustness of our framework, we provide two types: one type is MAOPT-GOA, which is adopted in fully cooperative settings (with access to global state and reward). The other type contains MAOPT-LOA and MAOPT-SRO, which are proposed for mixed settings (only access to local state with inconsistency and also individual rewards). MAOPT-SRO is proposed to solve the sample inconsistency due to the partial observation, by decoupling the dynamics of the environment from the rewards to learn the option-value function under each agent's preference. MAOPT can be easily combined with existing DRL approaches. Experimental results show it significantly boosts the performance of existing DRL methods. As for future work, it is worth investigating how to achieve coordination among agents by designing MAOPT-GOA in a centralized training, decentralized execution manner. For example, it is worth investigating how to decompose the joint option-value function into individual option-value functions and update each termination probability separately.

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

# A  PSEUDO-CODE FOR MAOPT-GOA AND MAOPT-IOA

---

**Algorithm 2** MAOPT-GOA.

---

1: Initialize: the joint option set $\Omega_1 \times \cdots, \times \Omega_n$, each $\Omega_i = \{\omega_1, \omega_2, \cdots, \omega_n\}$, joint option-value network parameters $\psi$, joint option-value target network parameters $\psi'$, termination network parameters $\varpi$, replay buffer $\mathcal{D}$, actor network parameters $\rho^i$ for each agent $i$, critic network parameters $\upsilon^i$ for each agent $i$, batch size $T$ for PPO
2: **for** each episode **do**
3:     Start from state $s$
4:     Select a joint option $\vec{\omega}$
5:     **for** each agent $i$ **do**
6:         Select an action $a^i \sim \pi^i(o^i)$
7:     **end for**
8:     Perform the joint action $\vec{a} = \{a^1, \cdots, a^n\}$
9:     Observe reward $\mathbf{r} = \{r^1, \cdots, r^n\}$ and new state $s'$
10:     Store transition $(s, \vec{a}, \mathbf{r}, \vec{\omega}, s')$ replay buffer $\mathcal{D}$
11:     Select another joint option $\vec{\omega}'$ if $\vec{\omega}$ terminates
12:     **for** every $T$ steps **do**
13:         **for** each agent $i$ **do**
14:             Set $\pi^i_{old} = \pi^i$
15:             Calculate the advantage $A^i = \sum_{t'>t} \gamma^{t'-t} r^i_t - V_{\upsilon^i}(o^i_t)$
16:             Optimize the critic loss $L^i_c = -\sum_{t=1}^{T} (\sum_{t'>t} \gamma^{t'-t} r^i_t - V_{\upsilon^i}(o^i_t))^2$
17:             The option advisor calculates the transfer loss $L^i_{tr} = f(t)H(\pi_{\omega_i}|\pi^i)$
18:             Optimize the actor loss $\bar{L}^i_a = \sum_{t=1}^{T} \frac{\pi^i(a^i_t|o^i_t)}{\pi^i_{old}(a^i_t|o^i_t)} A^i_t - \lambda KL[\pi^i_{old}|\pi^i] + L^i_{tr}$ w.r.t $\rho^i$
19:         **end for**
20:     **end for**
21:     Sample a batch of $B$ transitions from $\mathcal{D}$
22:     **for** each sample $(s, \vec{a}, \mathbf{r}, s')$ **do**
23:         **for** each $\vec{\omega}$ **do**
24:             **if** $\pi_{\omega_i}$ selects action $a^i$ for all $\omega_i \in \vec{\omega}$ **then**
25:                 Calculate $U(s', \vec{\omega}|\psi')$
26:                 Set $y \leftarrow \mathbf{r} + \gamma U(s', \vec{\omega}|\psi')$
27:                 Optimize the following loss w.r.t $\psi$: $L_{\vec{\omega}} = \frac{1}{B} \sum_b (y_b - Q_{\vec{\omega}}(s_b, \vec{\omega}|\psi))^2$
28:                 Optimize the termination loss w.r.t $\varpi = \varpi - \alpha_\varpi \frac{\partial \beta(s', \vec{\omega}|\varpi)}{\partial \varpi} A(s', \vec{\omega}|\psi') + \xi$
29:             **end if**
30:         **end for**
31:     **end for**
32:     Copy $\psi$ to $\psi'$ every $k$ steps
33: **end for**

---

Algorithm 2 illustrates the whole procedure of MAOPT-GOA. First, we initialize the network parameters for the joint option-value network, termination network, joint option target network, and the actor and critic networks of each agent $i$. For each episode, each agent $i$ first obtains its local observation $o^i$ which corresponds to the current state $s$ (Line 3). Then, MAOPT-GOA selects a joint option $\vec{\omega}$ for all agents (Line 4), and each agent selects an action $a^i$ following its policy $\pi^i$ (Lines 5-7). The joint action $\vec{a}$ is performed, then the reward $\mathbf{r}$ and new state $s'$ is returned from the environment (Lines 8, 9). The transition is stored in the replay buffer $\mathcal{D}$ (Line 10). If $\vec{\omega}$ terminates, then MAOPT-IOA selects another joint option $\vec{\omega}'$ (Line 11). For each update step, each agent updates its critic network by minimizing the loss $L^i_c$ (Line 16), and updates its actor network by minimizing the summation of the original loss $L^i_a$ and the transfer loss $L^i_{tr}$ (Line 18).

For the update of GOA, it first samples a batch of $B$ transitions from the replay buffer $\mathcal{D}$ (Line 21). Then GOA updates the joint option-value network by minimizing the standard L2 loss $L(\vec{\omega})$ (Lines 22-27). At last, the termination probability of the selection joint option is updated (Line 28).

---

**Algorithm 3** MAOPT-IOA.

---

1: Initialize: the option set $\{\omega_1, \cdots, \omega_n\}$, option-value network parameters $\psi$, option-value target network parameters $\psi'$, termination network parameters $\varpi$, replay buffer $\mathcal{D}^i$ for each agent $i$, actor network parameters $\rho^i$ for each agent $i$, critic network parameters $\upsilon^i$ for each agent $i$, batch size $T$ for PPO

2: **for** each episode **do**

3:     Start from state $s$

4:     **for** each agent $i$ **do**

5:         Select an option $\omega$

6:         Select an action $a^i \sim \pi^i(o^i)$

7:     **end for**

8:     Perform the joint action $\vec{a} = \{a^1, \cdots, a^n\}$

9:     Observe reward $\mathbf{r} = \{r^1, \cdots, r^n\}$ and new state $s'$

10:     **for** each agent $i$ **do**

11:         Store transition $(o^i, a^i, r^i, o^{i'}, \omega, i)$ replay buffer $\mathcal{D}^i$

12:         Select another option $\omega'$ if $\omega$ terminates

13:     **end for**

14:     **for** every $T$ steps **do**

15:         **for** each agent $i$ **do**

16:             Set $\pi^i_{old} = \pi^i$

17:             Calculate the advantage $A^i = \sum_{t'>t} \gamma^{t'-t} r^i_t - V_{\upsilon^i}(o^i_t)$ f

18:             Optimize the critic loss $L^i_c = - \sum_{t=1}^{T} (\sum_{t'>t} \gamma^{t'-t} r^i_t - V_{\upsilon^i}(o^i_t))^2$

19:             The option advisor calculates the transfer loss $L^i_{tr} = f(t)H(\pi_\omega|\pi^i)$

20:             Optimize the actor loss $\bar{L}^i_a = \sum_{t=1}^{T} \frac{\pi^i(a^i_t|o^i_t)}{\pi^i_{old}(a^i_t|o^i_t)} A^i_t - \lambda KL[\pi^i_{old}|\pi^i] + L^i_{tr}$ w.r.t $\rho^i$

21:         **end for**

22:     **end for**

23:     Sample a batch of $B/N$ transitions from each $\mathcal{D}^i$

24:     **for** each sample $(o^i, a^i, r^i, o^{i'})$ **do**

25:         **for** each $\omega$ **do**

26:             **if** $\pi_\omega$ selects $a_i$ at $o_i$ **then**

27:                 Calculate $U(o^{i'}, \omega|\psi')$

28:                 Set $y \leftarrow r^i + \gamma U(o^{i'}, \omega|\psi')$

29:                 Optimize the option-value network by minimizing the following loss w.r.t $\tau$:
$$L_\omega = \frac{1}{B} \sum_b \left(y_b - Q_\omega(o^i, \omega|\psi)\right)^2$$

30:                 Optimize the termination network w.r.t $\varpi$: $\varpi = \varpi - \alpha_\varpi \frac{\partial \beta(o^{i'}, \omega|\varpi)}{\partial \varpi} A(o^{i'}, \omega|\psi') + \xi$

31:             **end if**

32:         **end for**

33:     **end for**

34:     Copy $\psi$ to $\psi'$ every $k$ steps

35: **end for**

---

Algorithm 3 illustrates the whole procedure of MAOPT-IOA. First, we initialize the network parameters for the option-value network, termination network, option target network, and the actor and critic networks of each agent $i$. For each episode, each agent $i$ first obtains its local observation $o^i$ which corresponds to the current state $s$ (Line 3). Then, MAOPT-IOA selects an option $\omega$ for each agent $i$ (Line 5), and each agent selects an action $a^i$ following its policy $\pi^i$ (Line 6). The joint action $\vec{a}$ is performed, then the reward $\mathbf{r}$ and new state $s'$ is returned from the environment (Lines 8, 9). The transition is stored to each agent's replay buffer $\mathcal{D}^i$ (Line 11). If $\omega$ terminates, then MAOPT-IOA selects another option for agent $i$ (Line 12). For each update step, each agent updates its critic network by minimizing the loss $L^i_c$ (Line 18), and updates its actor network by minimizing the summation of the original loss $L^i_a$ and the transfer loss $L^i_{tr}$ (Line 20).

For the update of IOA, it first samples a batch of $B/N$ transitions from each agent's buffer $\mathcal{D}^i$ (Line 23). Then MAOPT-IOA updates the option-value network by minimizing the standard L2 loss $L(\omega)$ (Lines 24-29). At last, the termination probability of the selection option is updated (Line 30).

## B  ENVIRONMENT ILLUSTRATIONS AND DESCRIPTIONS

Pac-Man (van der Ouderaa, 2016) is a mixed cooperative-competitive maze game with one pac-man player and two ghost players (Figure 7). The most complex scenario is shown in Figure 10 with four ghost players and one pac-man player. The goal of the pac-man player is to eat as many pills (denoted as white circles in the grids) as possible and avoid the pursuit of ghost players. For ghost players, they aim to capture the pac-man player as soon as possible. In our settings, we aim to control the two ghost players and the pac-man player as the opponent is controlled by well pre-trained PPO policy. The game ends when one ghost catches the pac-man player or the episode exceeds 100 steps. Each player receives $-0.01$ penalty each step and $+5$ reward for catching the pac-man player.

MPE (Lowe et al., 2017) is a simple multiagent particle world with continuous observation and discrete action space. We evaluate the performance of MAOPT on two scenarios: predator and prey (Figure 8 with four agents and Figure 11 with twelve agents), and cooperative navigation (Figure 9 with four agents and Figure 12 with ten agents). The predator and prey contains three (nine in Figure 11) agents (green) which are slower and want to catch one adversary (blue)(rewarded $+10$ by each hit). The adversary is faster and wants to avoid being hit by the other three agents. Obstacles (grey) block the way. The cooperative navigation contains four (ten in Figure 11) agents (green), and four corresponding landmarks (cross). Agents are penalized with a reward of $-1$ if they collide with other agents. Thus, agents have to learn to cover all the landmarks while avoiding collisions. At each step, each agent receives a reward of the negative value of the distance between the nearest landmark and itself. Both games end when exceeding 100 steps.

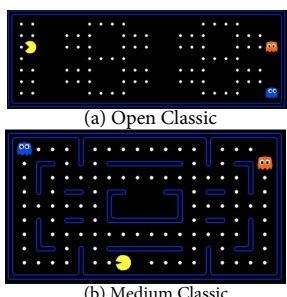

(a) Open Classic

(b) Medium Classic

Figure 7: Pac-Man.

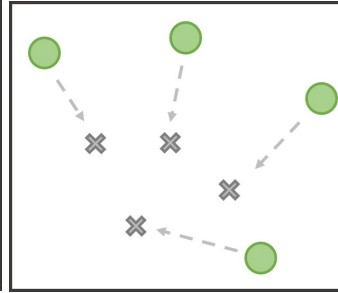

Figure 8: Predator and prey ($N = 4$).

Figure 9: Cooperative navigation ($N = 4$).

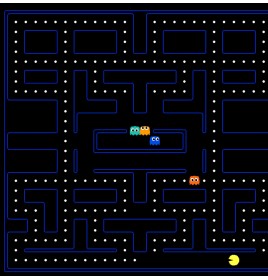

Figure 10: OriginalClassic.

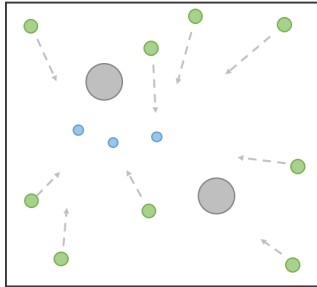

Figure 11: Predator and prey ($N = 12$).

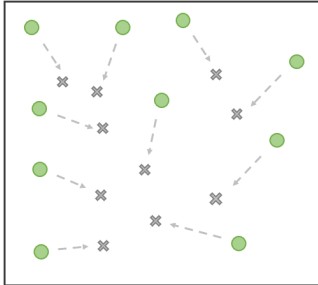

Figure 12: Cooperative navigation ($N = 10$).

**State Description**

**Pac-Man** The layout size of two scenarios are $25 \times 9$ (OpenClassic), $20 \times 11$ (MediumClassic) and $28 \times 27$ (OriginalClassic) respectively. The observation of each ghost player contains its position,

the position of its teammate, walls, pills, and the pac-man, which is encoded as a one-hot vector. The input of the network is a 68-dimension in OpenClassic, 62-dimension in MediumClassic and 111-dimension in OriginalClassic.

**MPE** The observation of each agent contains its velocity, position, and the relative distance between landmarks, blocks, and other agents, which is composed of 18-dimension in predator and prey with four agents (36-dimension with twelve agents), 24-dimension in cooperative navigation with four agents (60-dimension with ten agents) as the network input.

## C   ADDITIONAL RESULTS AND ANALYSIS

We here summarize the components and properties of our framework, and list the suitable scenarios of each option advisor. MAOPT-GOA contains $n$ agent models and 1 model for learning the joint-option value function and termination probabilities. MAOPT-GOA is used when we can obtain the global information. While in practice, only partial observations are available in some environments. Therefore, we also provide MAOPT-LOA to enable knowledge transfer among agents. MAOPT-GOA contains $n$ agent models and 1 model for learning the individual option value function and termination probabilities. The option model adopts the parameter sharing similar to common MARL training. However, each agent only obtains the local observation and individual reward signals, which may be different for different agents even at the same state. If we use inconsistent experiences to update the option-value network and termination network, the estimation of the option-value function would oscillate and become inaccurate. Due to partial observability and reward conflict, we design a novel option learning based on successor features. MAOPT-SRO contains $n$ agent models and 1 model for learning the individual SRO value function and termination probabilities. The SRO model adopts the parameter sharing similar to common MARL training.

Table 2: Aspects of MAOPT with three kinds of option adivsors.

|  | MAOPT-GOA | MAOPT-LOA | MAOPT-SRO |
|---|---|---|---|
| Components | $n$ agent's models 1 joint option model | $n$ agent's models 1 option model (parameter sharing) | $n$ agent's models 1 SRO model (parameter sharing) |
| Partial observation | — | ✓ | ✓ |
| Reward conflicts | ✓ | — | ✓ |
| The degree of flexibility | weak | strong | strong |

Figures 13 and 14 are the enlarged version corresponding to Figures 5 (a) and (b). Table 3 and 4 presents the average return achieved in predator and prey under two baselines: PPO and MADDPG.

Table 3: Average return with standard deviation in predator and prey ($N = 4$). First maximum value is bolded.

| PPO | DVM | MAOPT-GOA | MAOPT-LOA | MAOPT-SRO |
|---|---|---|---|---|
| $170.17\pm32.4$ | $241.87 \pm 12.38$ | $285.27 \pm 24.66$ | $268.67 \pm 23.18$ | $\mathbf{296.58\pm18.75}$ |

Table 4: Average return with standard deviation in predator and prey ($N = 4$). First maximum value is bolded.

| MADDPG | MAOPT-GOA | MAOPT-SRO |
|---|---|---|
| $229.68\pm6.21$ | $301.48 \pm 36.22$ | $\mathbf{338.8 \pm 8.78}$ |

Experimental results on games shown in Figure 10, 12 and 11 are shown in the following Figure 15. We can observe that MAOPT-SRO outperforms PPO and DVM, scales well with the increase in the number of agents.

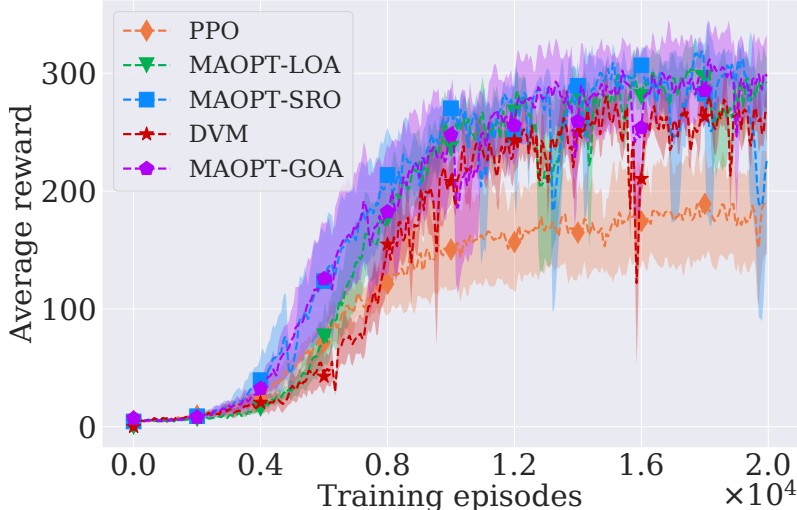

Figure 13: The performance on predator and prey ($N = 4$).

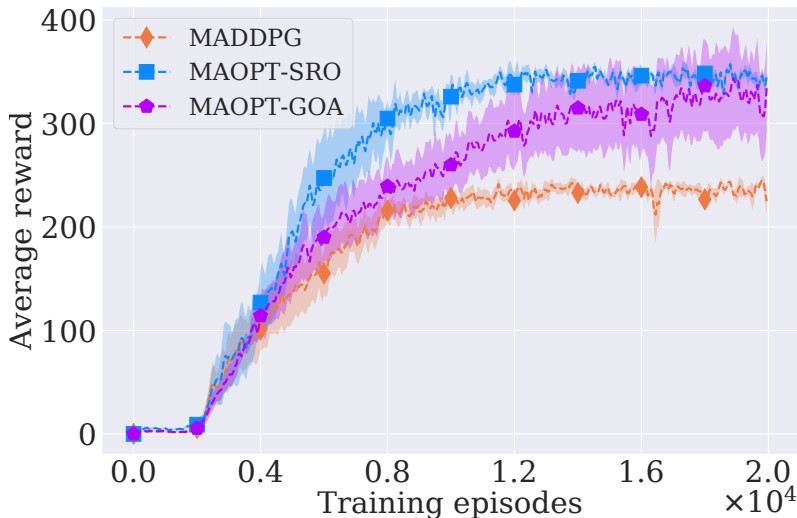

Figure 14: The performance on predator and prey ($N = 4$).

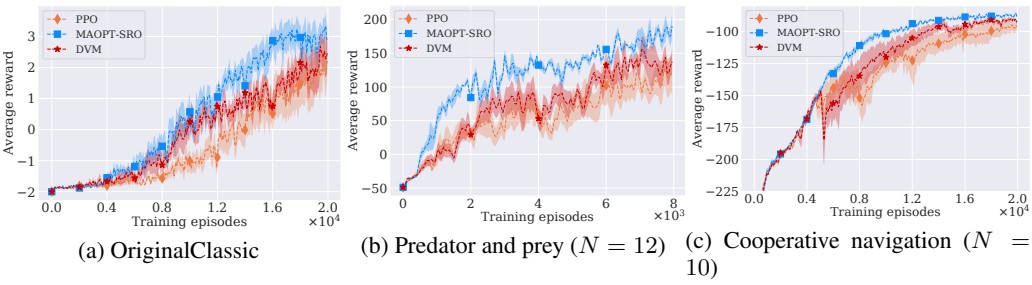

(a) OriginalClassic  (b) Predator and prey ($N = 12$)  (c) Cooperative navigation ($N = 10$)

Figure 15: The performance on various games.

# D   THEORETICAL ANALYSIS

We first explain that the advised policy is better than the agent's own policy. If none of other agents' policies is better than the agent's own policy, then the advised policy is the agent's own policy, which means there is no need to imitate.

The intuitive explanation of such transfer among agents is based on mutual imitation among agents. If an agent imitates a policy which is better than its own policy, it can achieve higher performance. Which policy should be imitated by which agent is decided by our option advisor. The option-value function estimates the performance of each option, as well as the intra-option policy, therefore we select the option with the maximum option-value for each agent to imitate the intra-option policy of this option. The convergence of option-value learning has been proved and verified (Sutton et al., 1999; Bacon et al., 2017). Therefore, the advised policy is the best among all policies at current timestep.

Then we provide the theoretical analysis to show the agent's policy will finally converge to the imitated policy through imitation. Give $n$ agents, $n$ options, for each agent $i$, the option advisors selects an option $\omega = \arg\max_{\Omega} Q_{\omega}(s, \omega)$ and $\omega$ contains the policy $\pi_{i,t}^*(\rho_t^*)$ with the maximum expected return $\eta_{i,t}^*(s)$. Each agent imitates the advised policy $\pi_{i,t}^*(\rho_t^*)$ to minimizes the difference between two policies $\pi_{i,t}^*(\rho_t^*)$ and $\pi_{i,t}(\rho_t^i)$: $\triangle \rho_i = \alpha_i(\eta_{i,t}^*(s|\rho_t^*) - \eta_{i,t}(s|\rho_t^i))(\rho_t^* - \rho_t^i)$. If we set $x_i = \rho_t^* - \rho_t^i$ then we calculate the difference of $x_i$ as follows: $\triangle x_i = -\alpha_i(\eta_{i,t}^*(s) - \eta_{i,t}(s))x_i$,

Then we have

$$\triangle \vec{x} = \begin{bmatrix} -\alpha_1(\eta_{1,t}^*(s|\rho_t^*) - \eta_{1,t}(s|\rho_t^1)) & 0 & \cdots & 0 \\ 0 & -\alpha_2(\eta_{2,t}^*(s|\rho_t^*) - \eta_{2,t}(s|\rho_t^2)) & \cdots & 0 \\ 0 & 0 & \cdots & 0 \\ 0 & 0 & \cdots & -\alpha_n(\eta_{n,t}^*(s|\rho_t^*) - \eta_{n,t}(s|\rho_t^n)) \end{bmatrix} \vec{x}$$

$$= A\vec{x}$$

Note that $\eta_{i,t}^*(s) - \eta_{i,t}(s) \geq 0$, then $-\alpha_i(\eta_{i,t}^*(s) - \eta_{i,t}(s)) \leq 0$, the main diagonal of the diagonal matrix $A$ only contains non-positive values. Therefore, the real part of all eigenvalues is non-positive. By means of Lyapunov's stability theorem (Shil'nikov, 2001), it is proved that $A$ is globally and asymptotically stable. The extreme of each $x_i$ approaches 0: $\lim_{t\to\infty} x_i = 0$ for $i \in \{1, 2, \cdots, n\}$. Therefore, each policy would converge to the advised policy through imitation.

To conclude, we show that for each agent, the advised policy is better than the policy of the agent itself, and each policy would converge to the advised policy through imitation. Thus, each agent's policy will converge to an improved policy through imitation, and this will not affect the convergence of the vanilla RL algorithm.

# E   NETWORK STRUCTURE AND PARAMETER SETTINGS

**Network Structure** Here we provide the network structure for PPO and MAOPT-SRO shown in Figure 16 (a) and (b) respectively.

**Parameter Settings**

Here we provide the hyperparameters for MAOPT, DVM as well as two baselines, PPO and MAD-DPG shown in Table 5 and 6 respectively.

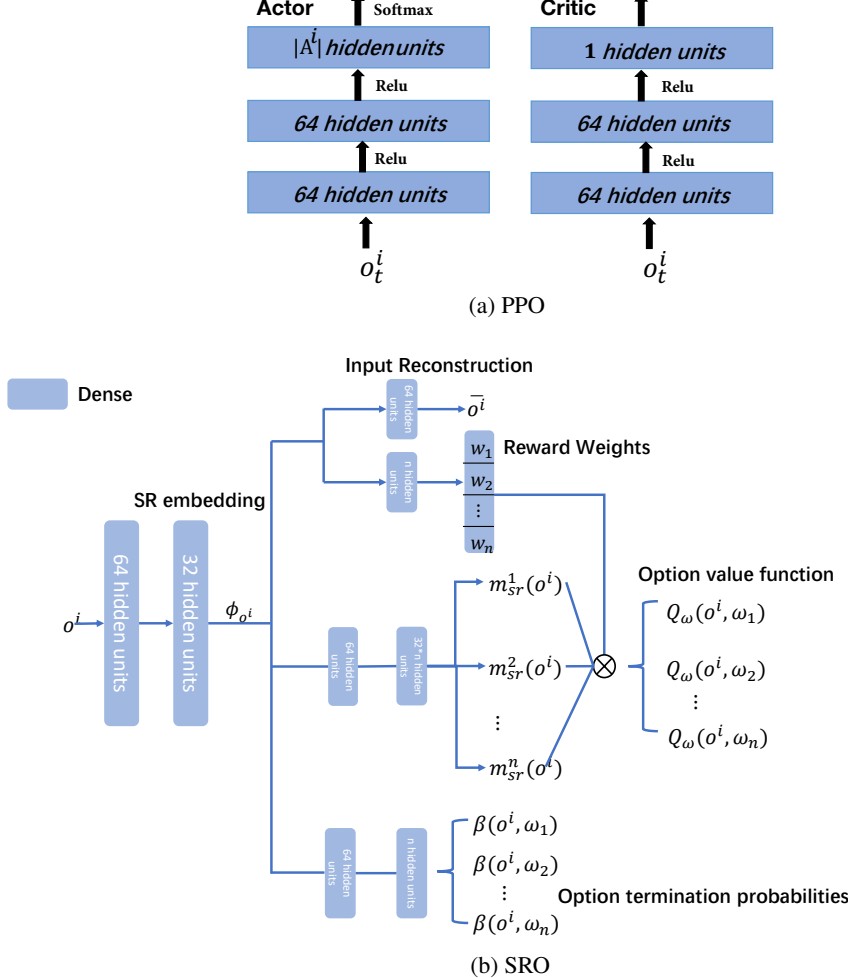

(a) PPO

(b) SRO

Figure 16: Network structures.

Table 5: Hyperparameters for all methods based on PPO.

| Hyperparameter | Value |
|---|---|
| Learning rate | $3e-4$ |
| Length of trajectory segment $T$ | 32 |
| Gradient norm clip $\lambda$ | 0.2 |
| Optimizer | Adam |
| Batch size $B$ of the option advisor | 32 |
| Replay memory size | $1e5$ |
| Learning rate | $1e-5$ |
| Action-selector $\epsilon$-start | $\epsilon$-greedy 1.0 |
| $\epsilon$-finish | 0.05 |
| $\epsilon$ anneal time | $5e4$ step |
| target-update-interval | 1000 |
| distillation-interval for DVM | $2e5$ step |
| distillation-iteration for DVM | 2048 step |

Table 6: Hyperparameters for all methods based on MADDPG.

| Hyperparameter | Value |
|---|---|
| Learning rate | $1e-2$ |
| Batch size | 1024 |
| Optimizer | Adam |
| Batch size $B$ of the option advisor | 32 |
| Replay memory size | $1e5$ |
| Learning rate | $1e-5$ |
| Action-selector $\epsilon$-start | $\epsilon$-greedy 1.0 |
| $\epsilon$-finish | 0.05 |
| $\epsilon$ anneal time | $5e4$ step |
| target-update-interval | 1000 |

