# OpenReview forum: "Transfer among Agents: An Efficient Multiagent Transfer Learning Framework"
_ICLR.cc/2021/Conference — Reject_

### Official Review · AnonReviewer4 · 2020-10-14
**Promising work but the manuscript needs polishing**

**Rating:** 6
**Confidence:** 3

**Review:**

---------------------
Post-rebuttal
---------------------
I am improving my grade a bit. I recommend the authors to dedicate some time further improving the paper clarity, especially in the matters related to my review and the other reviewers'

---------------------

The authors propose a transfer learning framework for transferring knowledge in multiagent tasks. Their method consists of learning a centralized option-based advisor, that will extract advice to provide to all agents in the system based on the options learned. This advice will be used to compute an auxiliary cost function that will ideally guide all agents towards learning faster.

---------------------
Pros
+ The proposal is novel as far as I can tell
+ The experimental evaluation shows very good performance in all evaluated scenarios
+ Timely and relevant research

----------------------------
Cons
- The manuscript needs polishing. Some English review is needed (paper is understandable but has many small mistakes)
- The assumptions of the method are not very clearly discussed.
- The novel parts of the paper are kind of intertwined with equations and ideas proposed in other works, which makes it harder to quickly see what was proposed in this specific work.
- I feel like LeCTR (Omidshafiei, 2019), cited in the paper, should have been included in the experimental evaluation

----------------------
 Suggestions
----------------------
- I am missing a clear and comprehensive discussion about the assumptions regarding the proposal. Specifically, which communication channels should be available to the method? Does the option advisor need to be able to communicate with all agents all the time? Anything changes regarding the needed infrastructure if we are using GOA, LOA, or SRO?

- Section 3 presents newly proposed concepts together with equations that have been introduced in the option-critic paper, for example. Section 3 should explain only new concepts and equations, making it clear what is a new proposal and what has been proposed before (that ideally should be described in the background section). Also, this section is a little verbose, you can make it more objective to improve readability.

- Why is DVM worse than the regular PPO in Figure 4 (a). It seems to me the algorithm is incorrectly configured or that the algorithm performs poorly in this specific scenario - in which case you should also evaluate another algorithm that performs better in this scenario. In general, I also think that you should add LeCTR at least in the evaluation in the particle environment.

- Add a table at the beginning of section 3 outlining the scenarios in which SRO, LOA, and GOA perform best, and which aspects affect in the algorithm choice.
-----------
minor
----------
- don't let titles without text (3. Approach)
- the paper needs an English review

---

> ### Author Response · Authors · 2020-11-19
> **We appreciate the reviewer’s inspiring comments.  We have clarified related contents in the revised version. We also provide clarifications for your other concerns.**
>
> We appreciate the reviewer’s inspiring comments. We have clarified related contents in the revised version. We also provide clarifications for your other concerns.
>
> **Q: About the assumptions**
>
> A: Thanks for your advice. We have rephrased this in the revised version. Our basic assumption is that knowledge transfer happens among homogeneous and cooperative agents, and good knowledge transfer could accelerate policy learning.
>
> **Q: About the communication**
>
> A: There is no communication between option advisors and agents during execution, and each agent makes decisions individually. During training, we assume the option advisor has access to the policies of agents to calculate the complementary loss function for each agent. The difference between GOA, LOA, and SRO lies in the information they can obtain. For GOA, it can obtain the global state, while for LOA and SRO, they can only obtain each agent’s local observation.
>
> **Q: About the structure**
>
> A: Thanks for your kind advice. We have re-organized section 3 in our revised version.
>
> **Q: DVM in fig 4(a)**
>
> A: The performance of DVM and PPO is competitive in Figure 4(a). The average return of DVM is $2.46$, and the average return of PPO is $2.53$, so the difference in the performance is not statistically significant. This is mainly due to the simplicity of the game, so we have conducted a more complex Pac-Man scenario (shown in Appendix C) to show the performance of our framework compared with previous work.
>
> **Q: About LeCTR**
>
> A: We do not compare with LeCTR because it only supports 2-agent scenarios. How to extend LeCTR to n-player games remains their future work. However, the particle environment contains at least 4 agents.
>
> **Q: Add a table**
>
> A: Thanks for your advice. We have clarified this in the revised version. Due to the page limit, we have added this in Appendix C.
>
> **Q: Minor**
>
> A: Thanks for your kind advice. We have polished our paper and modify the title of section 3.

---

### Official Review · AnonReviewer3 · 2020-10-22
**Efficient policy transfer method with option-based learning in multi-agent RL**

**Rating:** 6
**Confidence:** 3

**Review:**

##########################################################################

Summary:

The paper proposes a new option-based policy transfer framework for multi-agent reinforcement learning (MARL) called MAOPT. By framing multi-agent transfer as an option learning problem, MAOPT methods are able to learn when to give advice to agents and when to stop it. Authors provide a version of MOAPT for fully cooperative setting based on global state and reward, as well as two versions for mixed settings based on local states and per-agent rewards. The paper presents experimental results on two environments that show performance gains over existing RL methods.

##########################################################################

Reasons for score:

I vote for accepting the paper. I like the idea of utilising the option-based transfer learning approach applied in MARL. My concerns regarding the paper are mainly about the experiments. Hopefully, these would be addressed during the rebuttal period.

##########################################################################

Pros:

1. The proposed option-based method offers a new prospective for policy transfer in MARL. The design of the MAOPT is interesting and reasonable for many MARL problems.
2. The authors propose two different variants of the MAOPT that correspond to different MARL scenarios, namely fully-cooperative and mixed settings. The mixed setting is also studied by two different models, namely MAOPT-LOA and MAOPT-SRO, the latter of which is specifically designed to handle the experience inconsistencies of agents.
2. This paper provides results of several experiments, including both qualitative analysis and quantitative results, to show the effectiveness of the proposed framework.

##########################################################################

Cons:

1. It is interesting to see how the methods scale with the increase of the number of agents. To this end, it would have been great to see experimental results including more than a handful of agents. Even if the methods don't scale outstandingly well, it would be worth adding a discussing regarding this.
2. The authors mention two existing transfer learning methods, namely DVM and LTCR, but compare their method only with the former. It isn't clear to me why the authors haven't used LTCR as a baseline.

##########################################################################

Questions during rebuttal period:

Please address and clarify the cons above.

Also, I'm not really sure about this statement in the introduction: "in a multiagent system (MAS), the exploration strategy of each agent is different...". For instance, there is a MARL method called MAVEN that coordinates the exploration of all agents by conditioning their behaviour on joint variable from a latent space (http://papers.nips.cc/paper/8978-maven-multi-agent-variational-exploration). Maybe you could rephrase your statement to make it a bit more clear?

---

> ### Author Response · Authors · 2020-11-19
> **We appreciate the reviewer’s inspiring comments. We will provide additional experiments in Appendix C (to be added soon). We also provide clarifications for your other concerns.**
>
> We appreciate the reviewer’s inspiring comments. We will provide additional experiments in Appendix C (to be added soon). We also provide clarifications for your other concerns.
>
> **Q: Experiments on a large number of agents**
>
> A: AThanks for your advice. Our MAOPT supports a larger number of agents. We have conducted additional experiments on a more complex Pac-Man scenario with five agents and MPE with ten agents. We will plot the results and put them in Appendix C, page 16 as soon as possible.
>
> **Q: About LTCR**
>
> A: We think LTCR is not comparable under our settings due to the following reasons: 1) LTCR assumes that each agent exchanges its part of samples to another agent, 2) LTCR designs a specific communication protocol for the message exchange between agents. In our settings, each agent does not communicate with each other. It learns its own policy based on its own samples and the advice obtained from the option advisor.
>
> **Q: Description in the introduction**
>
> A: Thanks for your advice. We have clarified this in the revised version.

---

> > ### Author Response · Authors · 2020-11-25
> > **(Continued)**
> >
> > We have added new experimental results shown in Appendix C.

---

### Official Review · AnonReviewer2 · 2020-10-28
**An interesting extension to classic MARL model, but the design motivation of the mechanisms is not completely justified in my opinion**

**Rating:** 4
**Confidence:** 4

**Review:**


The authors propose a MARL solution based on the idea of "Local Option Advisor" and "Option-based Policy Transfer". The a


Strenghts

- This is timely area, in which it is good to see progress and alternative solutions to the existing ones.

- The problem of transfer learning might enable to apply MARL in different environments and application scenarios.

Weaknesses

- The significance of the proposed solution is unclear given the fact that the additional complexity of the approach proposed by the authors is not fully justified. Why is basic transfer learning not sufficient?

- The evaluation of the method is based on experiments, but the plots provided by the authors do not show definite evidence of the superior performance (the confidence interval are apparently overalapping).

- The aspects related to transfer learning are not fully evaluated in the paper.

In general, the actual contribution of this work with respect to the state of the art in MARL/transfer learning is not completely clear. The authors propose a rather complex solution, but its actual motivation is not apparent. In particular, it is unclear to see which particular design needs they are trying to address with these additional mechanisms. In other words, the advantages in adopting these mechanisms are not proven by the authors in my opinion.

Moreover, the method is evaluated experimentally, but its actual superiority is unclear. In fact, if you examine Figures 4 and 5, you see that the performance are comparable (the confidence appear to overlap).

Questions

- What are the situations where the proposed solution provides a real advantage? There is also a question about how the proposed solution generalizes to other problems: are there any underlying assumptions of the proposed additional mechanisms to a basic transfer learning model?

- Is there any specific theoretical evidence that shows that the addition of these mechanisms might lead to improved performance?

- What is the tradeoff of the proposed solution with respect to the added complexity? Is this complexity justified? Are there situations/problems that can be tackled only through these added mechanisms?

- Could you addditional explanations to the results in Table 1? How should we interpret them.

- Figure 6 is not completely "readable": how should we interpret the results presented in that figure?

---

> ### Author Response · Authors · 2020-11-19
> **We appreciate the reviewer’s inspiring comments. We have provided a theoretical analysis in Appendix D. We also provide clarifications for your other concerns.**
>
> We appreciate the reviewer’s inspiring comments. We have provided a theoretical analysis in Appendix D. We also provide clarifications for your other concerns.
>
> **Q: About the significance and the complexity**
>
> A: We here summarize the significance of our framework: 1) policy transfer base on option learning has been successfully applied in single-agent RL to handle the cases where each source policy is only partially useful for learning the target task [1,2]. We first leverage this idea in multiagent settings to enable knowledge transfer between agents in the same task. 2) Transfer among agents in the same task is meaningful because each agent's experience is different, if we figure out which part of the information is useful for each agent and transfer the knowledge, this will accelerate the whole training process and facilitate more efficient MARL. 3) To handle this problem, we model such transfer as the option learning. Meanwhile, this can handle the cases where an agent’s policy is only partially useful for another agent. We design three kinds of option advisors based on the assumptions of what information we can obtain during the training.
> 1.	MAOPT-GOA is used when we can obtain the global information. While in practice, only partial observations are available in some environments.
> 2.	Therefore, we also provide MAOPT-LOA to enable knowledge transfer among agents. However, in MAOPT-LOA, each agent only obtains the local observation and individual reward signals, which may be inconsistent for different agents even at the same state. If we use inconsistent experiences to update the option-value network and termination network, the estimation of the option-value function would oscillate and become inaccurate.
> 3.	Due to partial observability and reward conflict, we further design a novel option learning based on successor features (i.e., MAOPT-SRO) to handle these problems. Specifically, MAOPT-SRO decouples the dynamics of the environment from the rewards to learn the option-value function under each agent's preference, and experiments show MAOPT-SRO performs best on most of the test domains.
>
> Note that the network structure of MAOPT-GOA and MAOPT-LOA is the same, and they differ from the samples used for updates, so the complexity is the same for these two option advisors. The extra complexity lies in MAOPT-SRO since the SRO network architecture contains more parameters for updates. This is necessary because MAOPT-SRO can solve the reward conflicts and experiments show that MAOPT-SRO performs better than MAOPT-LOA and MAOPT-GOA in most of the test domains. The comparison among MAOPT-GOA, MAOPT-LOA, and MAOPT-SRO is shown in table 2, Appendix C. Thanks for pointing out this. We have clarified this in the revised version.
>
> [1] Context-aware policy reuse. AAMAS. 2019.
>
> [2] Efficient reinforcement learning via adaptive policy transfer. IJCAI. 2020.

---

> > ### Author Response · Authors · 2020-11-19
> > **(Continued)**
> >
> > **Q: Theoretical evidence**
> >
> > A: We first explain that the advised policy is better than or at least equal to the agent’s own policy. Then we provide the theoretical analysis to show the agent’s policy will finally converge to the advised policy through imitation. Combining these two evidences, it is easy to see that each agent’s policy will converge to an improved policy through imitation, and this will not affect the convergence of the vanilla RL algorithm. More details have been added in Appendix D. In the following, we show a simple illustration for these two evidences.
> > 1)	The advised policy is better than or at least equal to the agent’s own policy. The intuitive explanation of such transfer among agents is based on mutual imitation among agents. If an agent imitates a policy that is better than its own policy, it can achieve higher performance. Which policy should be imitated by which agent is decided by our option advisor. The option-value function estimates the performance of each option, as well as the intra-option policy, therefore we select the option with the maximum option-value for each agent to imitate the intra-option policy of this option. The convergence of option-value learning has been proved and verified [3,4]. Therefore, the advised policy is the best among all policies at the current timestep. Namely, the advised policy is better than or at least equal to the agent’s own policy. The equivalent case is that if none of other agents’ policies is better than the agent’s own policy, then the advised policy is the agent’s own policy, which means there is no need to imitate.
> > 2)	The agent’s policy will finally converge to the advised policy through imitation. Each agent imitates the advised policy to minimize the difference between the two policies (denoted as $x$). The advised policy performs better than the agent’s policy, thus it can obtain a larger return. If we calculate the difference equation of $x$, then we have a diagonal matrix with the main diagonal containing non-positive values only. Therefore, the real part of all eigenvalues is non-positive. By means of Lyapunov’s stability theorem [5], it is proved globally and asymptotically stable. Then the extreme of $x$ approaches $0$. Therefore, each policy would converge to the advised policy through imitation.
> >
> > To conclude, for each agent, we show that the advised policy is better than the policy of the agent itself, and each policy would converge to the advised policy through imitation. Therefore, it is easy to see that each agent’s policy will converge to an improved policy through imitation. More details have been added in Appendix D.
> >
> > [3] Between {MDP}s and semi-{MDP}s: A framework for temporal abstraction in reinforcement learning. Artificial intelligence. 1999.
> >
> > [4] Option-critic architecture. AAAI. 2017.
> >
> > [5] Methods of qualitative theory in nonlinear dynamics (Vol. 5). World Scientific. 2001.
> >
> > **Q: About the overlap**
> >
> > A: Thanks for pointing out this. We have added more experiments and analyses in our revised version. The overlap in figure 4(a) is due to the simplicity of the game, so we have evaluated our method on a more complex Pac-Man scenario (shown in Appendix C) to show the performance of our framework compared with previous work.
> > The performance overlap in figure 5(a) is due to the scale of y-axis, so it is hard to show the advantage of our framework. We have added two tables of the average return and a large-scale figure 5(a) in Appendix C.
> >
> > **Q: Explanations of Table 1**
> >
> > A: In the game of cooperative navigation, $n$ agents are required to cover $n$ landmarks while avoiding collisions. Therefore, a better result means to get a closer average distance between agents and landmarks, and fewer collision frequencies among agents. Following MADDPG, we provide the average distance between agents and landmarks (line 1 in table 1), and the average collisions among agents (line 2 in table 1) of all methods. For example, the average distance between agents and landmarks for PPO is 1.802, and the average collision frequency is 0.163. Thanks for pointing out this. We have rephrased this in our revised version.

---

> > > ### Author Response · Authors · 2020-11-19
> > > **(Continued)**
> > >
> > > **Q: Explanations of Figure 6**
> > >
> > > A: Thanks for pointing out this. We have rephrased this in our revised version. Each arrow in figure 6 is the direction of movement caused by the specific action at each location. So the four sub-figures show the direction of movement caused by the action selected from the policy of agent $i$ at $t_1=6000$th episode (figure 6a, top left), and at $t_2=20000$th episode (figure 6c, bottom left); the direction of movement caused by the action selected from the intra-option policies of MAOPT-SRO at $t_1=6000$th episode (figure 6b, top right), and at $t_2=20000$th episodes (figure 6d, bottom right) respectively. The preferred direction of movement should be towards the blue circle, and we can see that figure 6b contains more preferred movements than figure 6a, so does figure 6d compared with figure 6c. This means the option advisor advises a better policy to the agent than the agent’s own policy, thus agents can learn faster and achieve better performance than learning from scratch.
> > >
> > > We hope that these comments have addressed the reviewer’s concerns about the paper. We are happy to answer any follow-up questions.

---

### Official Review · AnonReviewer1 · 2020-10-30
**Interesting idea, but the some description and settings are confusing**

**Rating:** 6
**Confidence:** 5

**Review:**

This paper proposed an option-based framework for multiple agents to share knowledge with each other in the same MARL task. For scalability and robustness, two variants of the framework are designed, including 1) a global option advisor, which has the access to the global information of the environment; 2) local option advisor combined with successor representation option to enable more accurate option-value estimation. Experimental results demonstrate the proposed method is able to improve the performance of existing deep RL approaches for multiagent domains.


Pros:

1. This work models the policy transfer among agents as an option learning problem. This is an interesting idea for knowledge transfer in MARL tasks.

2. Comparing to previous teacher-student framework and policy distillation framework, a key feature is that the proposed framework is more adaptive (in what sense?) and applicable to scenarios consisting of more than two agents.


Cons:

1. Some details in are not clear (see Questions and Comments)

2. This framework based on centralized information, which may not always be available for multiagent problems, especially for noncooperative settings.

Questions and Comments:
The authors mention that “selecting a joint option means each advice given to each agent begins and ends simultaneously”. Does this mean the options (advice given) all have the same length? Especially for MAOPT-GOA, the termination function is over joint options. But this seems to be a very restricted assumption, given there are a lot of uncertainty in real world and different sub tasks may take different amount of time.

Advice is given in option level, level action selection are based on the intra-option policy, which is learned through imitation learning, does that mean the student need to collected the demonstration data from an expert (teacher?), how do you obtain the expert policy?

Are the agents homogenous? Meaning do they share the same option sets? How many options are available for each agent for the tasks in the experiment?

It is mentioned that “each option \omega^i contains an intra-option policy corresponding to an agent’s policy \pi^i”. It is unclear why the authors make this statement. Since in general \pi^i may not corresponds to the intra-option policy of \omega^i, unless there is only one option.

=======================After the rebuttal ========================
After rebuttal, I think the author have addressed most my questions. However, I agree with other reviewers’ opinions that the paper need further polishing and clarification, especially regarding the cooperative settings of the problems (homogeneous team, number of options etc as admitted by the authors) and associated theoretical and practical issues. Given the novelty of the idea and the current status of paper, I maintain my score.

---

> ### Author Response · Authors · 2020-11-19
> **We thank the reviewer for their valuable and insightful feedback. We have clarified related contents in the revised version.**
>
> We thank the reviewer for their valuable and insightful feedback. We have clarified related contents in the revised version.
>
> **Q: About MAOPT**
>
> A: Only MAOPT-GOA selects a joint option to use until termination and then selects another joint option, thus each agent’s advice begins and ends simultaneously. Note that this mechanism is not very flexible because MAOPT-GOA cannot handle the case where some agents may need to imitate their teachers for a longer time (which is also described in the last paragraph in section 3.2). Therefore, we also provide other two kinds of option advisors: LOA and SRO both of them use local observations and control each advice when to terminate individually. Thanks for pointing out this. We have rephrased this to make it clearer in the revised version.
>
> **Q: About the policy imitation**
>
> A: The student does not need to collect the demonstration data. During training, each agent’s complementary loss function (imitating the advised policy) is given by the option advisor, we assume the option advisor has access to agents’ policies during the training. Thanks for pointing out this. We have clarified this to make it clearer in the revised version.
>
> **Q: Homogenous**
>
> A: In our experimental settings, we assume the agents using the option advisor are homogenous, i.e., agents share the same option set. Please note that our MAOPT can also support the situation where each agent is initialized with different numbers of options e.g., each agent only needs to imitate its neighbors. To achieve this, instead of input states into the option-value network, we just input the pair of states and options to the network and output a single option value. Thanks for pointing out this. We have clarified this in the revised version.
>
> **Q: About the number of options**
>
> A: For a MAS with $n$ agents, we initialize $n$ options for each agent. Specifically, for agent $i$, the option set is $\Omega_i = ${$\omega_1, \omega_2,… \omega_n$}, and each option $\omega_i$ contains agent $i$’s policy $\pi_i$. As for GOA, the joint option set is the multiplication of each agent’s option set: $\Omega_1 \times \Omega_2 \times … \times \Omega_n$. Thanks for pointing out this. We have clarified this in the revised version to make it clearer.
>
> We hope that these comments have addressed the reviewer’s concerns about the paper. We are happy to answer any follow-up questions.

---

### Official Review · AnonReviewer5 · 2020-11-04
**A combination of distant fields, but difficult to understand**

**Rating:** 6
**Confidence:** 3

**Review:**

This paper considers the multi-agent Reinforcement Learning setting, and proposes a scalable method for the agents to help each other learn, by transferring their policies to each other. The main idea presented of the paper is that an agent i may optimize its policy $\pi_i$ using a combined loss, that not only optimizes the agent returns obtained by the agent, but also imitates the policy of some other agent j. Which agent should imitate what agent is also learned, using an approach inspired from the Options framework, and Option learning algorithms such as the Option-Critic architecture. Several variants of the proposed approach are introduced, depending on how much information the agents can communicate in a particular setting.

This paper is very interesting, as it proposes a method that combines two domains of Reinforcement Learning, namely Multi-Agent systems and Options, that are not often combined. The empirical results are encouraging, and the method seems to scale to a significant amount of agents.

However, the paper is, in my opinion, very difficult to fully understand. I'm an well versed in Options, and knowledgable in multi-agent systems, but the difficulty of understanding the paper does not come from a lack of background information on these two domains (successor features are also very well presented). In my opinion, small but important details are missing. For instance, a precise and explicit description of what the agents exchange as information (policy parameters, actions?), and when, would have helped. Figures 1 and 2, and most of the formulas of the paper, focus on the mathematical aspect of the proposed architecture. They allow to see that all the pieces of information fit together into sound formulas. But it is very difficult to see how to implement the algorithm in practice. I would suggest that the authors add pseudocode sections, for the acting loops of the agents, their training loops, how agents decide which agent to imitate, and how the option and termination functions are learned.

Because the ideas proposed in the paper are interesting, and seem original, I would recommend accepting this paper, but only if the authors manage to make the paper clearer for a first reader.

Author response: discussion with the authors allowed to clarify what the agent see and how they exchange information. I still find the paper a bit difficult to understand, most probably due to its novelty, but I consider that the current version is acceptable. As such, I recommend accepting this paper.

---

> ### Author Response · Authors · 2020-11-19
> **We thank the reviewer for the valuable and insightful feedback. We have clarified related contents in the revised version.**
>
> We thank the reviewer for their valuable and insightful feedback. We have clarified related contents in the revised version.
>
> **Q: exchange information**
>
> A: Each agent does not exchange information with each other directly, it only receives the complementary loss function from the option advisor during the training phase (i.e., each agent does not know which policy it imitates and how the extra loss function is calculated.). Each agent makes decisions individually during execution. Thanks for your advice. We have clarified this in the revised version. To make it clear, we have re-designed Figures 1 and 2 and rephrased the descriptions.
>
> **Q: add pseudocode**
>
> A: Thanks for your advice. Due to the page limit, we put the pseudocode in the appendix. We have clarified this and moved the pseudocode of MAOPT-SRO to the main context in the revised version. We also provide the pseudocode of MAOPT-GOA and MAOPT-LOA in appendix A (page 12, 13).

---

> > ### Comment · AnonReviewer5 · 2020-11-19
> > **Nice improvement of the clarity of the paper**
> >
> > Thank you for the revised paper, I think that the clarity largely improved. The remark about the fact that the agents receive losses, and do not know which policy they imitate, is also crucial to the understanding of the paper (I did not find it in the revised paper, did I miss something?).
> >
> > Algorithm 1 really helps understanding what happens in the proposed approach. I however still have a few questions, that I hope can help you identify the points of the algorithms that readers not familiar with it find difficult to understand:
> >
> > - Line 5 (in the algorithm): $\omega$ is selected, but that $\omega$ does not seem to be used before line 12 where it is replaced by another $\omega$.
> > - Line 19: $L^i_{tr}$ is not defined. Because I don't see $\omega$ used anywhere when training the agent, I suppose that $L^i_{tr}$ depends on $\omega$ in some way, and would be the imitation loss that the agent receives? (note: Section 3.3 explains what $L^i_{tr}$ is, I missed it at first, maybe a See Section 3.3 could be added to the pseudocode?)

---

> > > ### Author Response · Authors · 2020-11-19
> > > **Thanks for your comments.**
> > >
> > > **About the remark**
> > >
> > > A: We have clarified this at the beginning of Section 3.1 (colored by orange).
> > >
> > > **About algorithm 1**
> > >
> > > A: We are sorry that there is some misunderstanding of how $\omega$ is used. The first loop is the execution loop, line 12 is to check whether to terminate the option at the next state $s'$. While for the training loop, given each sample $(o^i, a^i, r^i, {o^{i}}^{'},\omega,i)$, we use the option advisor to calculate $L_{tr}^i=f(t)H(\pi_{\omega} |\pi_i)$, where $\omega$ is selected during execution. We are sorry to leave $\omega$ behind in the sample. We have clarified this in the revised version.
> > >
> > > About $L_{tr}^i$, we have clarified in algorithm 1 to make it clearer.
> > >
> > > We thank the reviewer's comments, this helps us a lot to improve the paper!

---

### Author Response · Authors · 2020-11-19
**Overall response about the revision**

We thank all the reviewers for their valuable and insightful feedback. After reading the comments, we do agree with the reviewers, and our paper misses important details about the algorithms and experiments, leading to some misunderstandings and deviation on the main points of this paper to be evaluated.

As a remedy, we upload a revision with the appendix, containing pseudo-code (section 3 and appendix A), further discussions and additional experiments (appendix B and C), the theoretical analysis (appendix D). Some parts in the main content are also further clarified to describe the details. All incremental and modified contents in this revision are colored by orange.

In the following, we try to address reviewers’ individual concerns and questions.

---

### Decision · Program_Chairs · 2021-01-07
**Final Decision**

**Decision:**

Reject

**Comment:**

The paper proposes a method on multi-agent options-based policy transfer where agents help each other learn by exchanging policies.

The core idea behind the paper is novel, as it addresses a new and emerging topic of social learning, and of interest to ICLR community. The authors significantly improved the paper with additional experiments and theoretical analysis during the rebuttal process, resulting in a compelling case for the significance of the method.

Unfortunately, the paper requires addressing the clarity, and a careful proofreading pass, making it unsuitable to ICLR in its current form,